

# YAP1 affects the prognosis through the regulation of stemness in endometrial cancer

Wei Kong[*], Yuzhen Huang[*], Peng Jiang, Yuan Tu, Ning Li, Jinyu Wang, Qian Zhou, Yunfeng Zheng, Shikai Gou, Chenfan Tian and Rui Yuan

Department of Gynecology, The First Affiliated Hospital of Chongqing Medical University, Chongqing, China
[*] These authors contributed equally to this work.

## ABSTRACT

**Background**. Endometrial cancer stem-like cells (ECSCs) have been proven to be responsible for recurrence, metastasis, and drug-resistance in patients with endometrial cancer. The HIPPO pathway has been shown to play an important role in the development and maintenance of stemness in a variety of tumors. While there was less research about its function in ECSCs. The aim of this study was to explore the role of YAP1, a core molecular of HIPPO pathway, in the stemness of endometrial cancer and to reveal its influence on prognosis.

**Methods**. We collected specimens and clinical data from 774 patients with endometrial cancer to analyze the correlation between YAP1 expression and prognosis. We then examined the expression of YAP1 in ECSCs and EC cell lines (Ishikawa; HEC1-A) in vitro experiments. Changes in the stemness of cell lines were detected after YAP1 silencing by siRNA. Finally, high-throughput sequencing was used to predict the potential molecular interactions and mechanisms of YAP1's effect on stemness.

**Result**. Down-regulation of YAP1 significantly suppresses the stemness of EC cell lines. High expression of YAP1 leads to poor prognosis in EC by regulation of stemness.

**Conclusion**. YAP1 plays an important role in the prognosis of patients with EC by regulation of stemness.

## INTRODUCTION

Endometrial cancer (EC) is a common gynecological cancer with a rising incidence, especially in high-income countries (*Lu & Broaddus, 2020*). An increase in the aged population, obesity, and other metabolite diseases may contribute to its high incidence (*Ferlay et al., 2018*). Many patients with EC were diagnosed at an early stage, and with appropriate treatment good survival was achieved (*Brooks et al., 2019*). However, postoperative recurrence and metastasis greatly influence the survival of patients with advanced endometrial cancer and remain great challenges for surgeons. Meanwhile, the molecular mechanisms of postoperative recurrence and metastasis of EC were still unclear. Recent research has proposed the concept of cancer stem-like cells (CSCs) in EC (*Giannone et al., 2019*), which contributes to specific self-renewal and differentiation

Corresponding author
Rui Yuan,
yrui96@hospital.cqmu.edu.cn

in cancer cells. Cancer stem-like cells' features and role in progression, invasiveness, and metastasis were identified (*Babaei, Aziz & Jaghi, 2021*; *Shin & Cheong, 2019*). CD44 was related to EC stemness reported by another research (*Elbasateeny et al., 2016*; *Park, Hong & Park, 2019*). Also, spheres generated from EC cell lines showed the co-expression of CD44 and CD133 in ECSCs (*Mirantes et al., 2013*; *Rutella et al., 2009*). In addition, several studies have linked NANOG to ECSCs (*Zhou et al., 2011*). OCT4 and SOX2 were reported to have great impacts on the deregulation of NANOG and were also found to be associated with the self-renewal capacity in ECSCs (*Hubbard et al., 2009*; *Zhou et al., 2011*). Also, the tumor spheres with a higher proliferation potential than differentiated cells showed a co-expression of NONAG, OCT4, and SOX2 mRNA (*Zhou et al., 2011*). Thus, the proteins including OCT4, SOX2, NANOG, CD44, and CD133 had been widely defined as stemness-related markers (*Mirantes et al., 2013*; *Rutella et al., 2009*; *Szymonik et al., 2021*). It is important to conduct further research on ECSCs because it would provide new insight into the understanding of recurrence and metastasis in EC.

Studies had illustrated that several pathways, such as the Notch signaling pathway, Wnt pathway, PI3K/AKT pathway, TGF-beta signaling, and the HIPPO signaling pathway, are involved in the process of stemness. Research showed that activation of the HIPPO pathway would lead to chemotherapy resistance and increasing of stemness in ovarian cancer (*Muñoz Galván et al., 2020*). A similar study also identified PFKFB3 as a novel target to regulate stem cells in small lung carcinoma through HIPPO pathway (*Thirusangu et al., 2022*). It has been proven that MCM2 promotes the stemness and sorafenib resistance of hepatocellular carcinoma cells *via* hippo signaling (*Zhou et al., 2022*). Also, by regulating the HIPPO pathway, exosomal miR-4800-3p influenced the stemness in hepatocellular carcinoma (*Lin et al., 2022*). YAP1, as a key role in the pathway, was also identified as a regulator of stemness in glioblastoma (*Castellan et al., 2021*). Researchers also claimed similar findings in colorectal cancer (*Tian et al., 2020*), ovarian cancer (*Li et al., 2020*), and osteosarcoma (*Shi, Li & Guo, 2021*), *etc*. Nowadays, few studies are focusing on the role of the HIPPO pathway in ECSCs. We still have not revealed the link between YAP1 and stemness in EC.

Therefore, our study aimed to investigate the effect of YAP1 on the prognosis of patients with EC and the function of YAP1 on the stemness of EC cell lines. Our study provides new insight into the study of the HIPPO pathway in EC stemness and reach the goal of finding the potential biological target for the recurrence and metastasis of EC.

## MATERIALS & METHODS

All procedures were performed by relevant guidelines and were in line with the World Medical Association Declaration of Helsinki. This study has been approved by the Ethics Committee of the First Affiliated Hospital of Chongqing Medical University (2021-676).

### Cases and follow-up

A total of 774 patients from 2015 to June 2020 who accepted primary surgery at the First Affiliated Hospital of Chongqing Medical University were enrolled in this study. Tissue samples were obtained from postoperative specimens after the definitive surgery, with

at least five cm of the tumor margin. The clinical data of cases should include at least the followings: age, BMI, FIGO staging, histotype, cervical stromal invasion, myometrial invasion, lymph vascular invasion (LVSI), *etc.* The procedure was approved by the Ethics Committee of the First Affiliated Hospital of Chongqing Medical University.

The postoperative follow-up was mainly carried out by outpatient services and telephone. The follow-up frequency was as follows: every 3 months for the first two years, then every half of a year for the following three years, and once a year after five years. The content of the follow-up mainly contained a physical examination, necessary imaging examination, and biochemical examination. Recurrence was confirmed by physical examination and imaging examination, including X-ray, CT, PET/CT, MRI and so on (*Ouldamer et al., 2016*). Patient survival was confirmed by telephone interviews. Recurrence-free survival (RFS) is defined as the time between the date of surgery and the date of recurrence (proved by imaging or histology) (*Huijgens & Mertens, 2013*). Death is defined as death due to postoperative recurrence. Overall survival (OS) is defined as the time between the date of surgery and the patient's date of death (*Ouldamer et al., 2018*).

## Immunohistochemistry and evaluation

The specimens were sent to the Department of Pathology at Chongqing Medical University (CQMU) for further processing. All the procedures were conducted by the same standards (*Jiang et al., 2020*). Paraffin was used to embed the postoperative specimens. The tissues would be made into 2–3 micrometers slices. H&E staining was used to identify the most cancerous part of the specimens. The histological parameters were recorded including histological type, size of lesion, grade, and invasion depth. These were first judged by the junior pathologists. Immunohistochemistry (IHC) was performed by an automated Immuno-Stainer (manufactured by Leica Bond-Max) and followed the instructions (*Yang et al., 2016*). The slices were dried for 12 h at a temperature of 60 °C. After deparaffinized, the slices were hydrated in 70% methanol for 10 min. Antigen retrieval was carried out by microwaved heating for 5 min in 0.1 mol/l citrate buffer. Endogenous peroxide was blocked by 3% $H_2O_2$ in methanol. After that, the slices were incubated with YAP1 antibody (1:400, 14074, Cell Signaling Technology), ER (SP1, in 1:50), PR (MX009, in 1:500), Ki67 (MX006, in 1:300), and P53 (MX008, in 1:200) (Maixin Biotech, Fuzhou, China) for 1 h, and then incubated with anti-rabbit secondary antibody (manufactured by Leica) at a room temperature for 30 min. 3,3′-diaminobenzidine tetrahydrochloride (DAB Substrate System, DAKO) and hematoxylin were used to color the slices.

Referring to other studies, we set the cutoff value of four factors as follows: ER $\alpha$, 5%; PR, 5%; Ki67, 40% (*Jia et al., 2021*; *Jiang et al., 2021*; *Jiang et al., 2020*). P53 was classified as normal or abnormal. On this basis, we defined the patient as having high or low expression. The evaluation of staining was as follows (*Jiang et al., 2021*): briefly, the average percentage of positive-stained cells in five random scopes was recorded. Slices with more than 50% positive-stained cells (in nuclear) were considered as high expression. Research has identified that nuclear localization of YAP1 was the activated form of YAP1 which would induce the subsequent mechanism and thus we focused on the nuclear staining of YAP1 (*Zanconato et al., 2015*). The frequency of YAP1 staining in nuclear was graded

from 0 to 4 by the percentage of positive cells as follows: grade 0, <3%; grade 1, 3–25%; grade 2, 25–50%; grade 3, 50–75%; grade 4, >75%. Grades 0, 1, and 2 were considered low expression, and grades 3 and 4 were high expression (*Ballout et al., 2022*). Two pathologists would evaluate the same slice (blind). If the results differed by less than 10%, the average of the two results was considered the ultimate result. Otherwise, they would reevaluate the slice and then reach a consensus (unblind).

## Cell culture and transfection

Ishikawa, HEC1-A, and RL95-2 were purchased from Cellbook Technology (Guangzhou, China). Short tandem repeat (STR) analysis was performed in all cell lines. Ishikawa and HEC1-A were cultured in DMEM (Gibco) supplemented with 10% FBS while RL95-2 was cultured in DMEM/F12 (Gibco) with 10% FBS.

ECSCs were grown in DMEM/F12 (Gibco) with 10 ng/ml hFGF (Peperotech), 20 ng/ml hEGF (Peperotech), and B27 supplement (17504044; Gibco) (*Chen et al., 2020*). 0.5 ml of all the above mediums was added to ECSCs cultured plates every other day. All the cell lines were grown at 37 °C in a 95% humidified atmosphere and 5% $CO_2$. And the ECSCs obtained from Ishikawa and HEC1-A were described as ECSCs$^{Ishikawa}$ and ECSCs$^{HEC1-A}$, respectively.

siRNA targeting human YAP1 mRNA (siRNA#1, siRNA#2, which are shown in File S1) and a non-specific scramble siRNA sequence (Control, which was shown in File S1) were synthesized (Genepharma, Shanghai, China) and then transfected into Ishikawa and HEC1-A using Lipofectamine 2000 (Invitrogen, Carlsbad, CA, USA) according to the manufacturer's instruction. Assays were performed after 2 days, which confirmed the efficiency of transfection.

## ECSCs sphere formation

After confirming the transfection of siRNA targeting human YAP1 in Ishikawa and HEC1-A, cells were followingly cultured in DMEM/F12 (Gibco) with 10 ng/ml hFGF (Peperotech), 20 ng/ml hEGF (Peperotech) and B27 supplement (17504044; Gibco) in six wells plates respectively, with a density of 20,000 cells for each well. The differences in sphere formation between siRNA#1,2 and siNC were compared on the 7th day. Before observation, the plate was slightly shaked so that the stem cell spheres in each well were distributed as evenly as possible in each field of view, and three central fields of view in each well were randomly observed under 100X field of view, photographed, and the number of stem cell spheres with maximum diameter >50 μm in each field of view was recorded using NIS-Elements BR software (version 4.60.00). The number of spheroids in each group was then counted in three random fields of view under the light microscope for statistical analysis. The average number represents the result. Each sample was counted three times (*Suzuki et al., 2021*).

## RNA extraction and quantitative reverse transcription PCR

Total RNA was extracted by using TRIzol® RNA Isolation Reagents (Invitrogen, Carlsbad, CA, USA). Quantitative reverse transcription PCR (qRT-PCR) was used to detect the difference in mRNA expressions. mRNA analysis was performed in Bio-Rad CFX Connect

Real-time PCR Instrument by using SYBR green master mix (Takara, Dalian, China). All used primers are shown in File S2. The PCRs were performed at least three times independently for each experiment. The relative expression of indicated genes compared with $\beta$-actin was calculated using the $2^{-\Delta\Delta CT}$ method.

## Western blotting assays

Western blotting assays were used to identify the different expressions of protein. Modified RIPA lysis buffer was used to collect the total cell lysates. BCA Protein Assay Kit (Beyotime, Jiangsu, China) was used to standardize the concentration of different samples. Proteins were separated by SDS-PAGE, transferred to a polyvinylidene fluoride (PVDF) membrane by using Mini-PROTEAN® Tetra (Bio-Rad, Hercules, CA, USA), and probed with primary antibodies respectively. The primary antibodies were as follows: YAP (D8H1X) XP® Rabbit mAb (1:1000, 14074S, Cell signaling technology), SOX2 Polyclonal antibody (1:1000, 11064-1-AP; Proteintech), OCT4 Polyclonal antibody (1:1000, 11263-1-AP; Proteintech), NANOG Polyclonal antibody (1:1000, 14295-1-AP; Proteintech), CD44 Polyclonal antibody (1:2000, 15675-1-AP; Proteintech), CD133 Polyclonal antibody (1:2000, 18470-1-AP; Proteintech), GAPDH Monoclonal antibody (1:50000, 60004-1-lg; Proteintech), Alpha Tubulin Polyclonal antibody (1:2000, 11224-1-AP; Proteintech). Subsequently the membranes were incubated with corresponding secondary antibodies: HRP-Goat anti-rabbit IgG (H+L) (RS0002; Immunoway); HRP-Goat anti-mouse IgG (H+L) (RS0001; Immunoway). Blots were detected using an enhanced chemiluminescence system (S6, manufactured by CLiNX, Shanghai, China).

## Online database

Kaplan–Meier Plotter (KM Plotter; http://kmplot.com/analysis/) is a website that can assess the correlation between all genes (mRNA) and survival in pan-cancer (*Lánczky & Győrffy, 2021*). STRING (Version 11.5; https://cn.string-db.org/) is a database that can show the protein–protein interaction (PPI) networks, which are based on current studies (*Szklarczyk et al., 2021*).

## High-throughput sequencing of the transcriptome

The ECSCs[Ishikawa] that we cultured were collected on the eighth day. High-throughput sequencing was conducted by LC-Biotechnology (Hangzhou, China). The detailed protocols and data analysis were recorded in File S3.

## Statistics

The differences in clinical features between the two groups as well as the univariate and multivariate analyses were performed in SPSS 26.0. The column graphs were plotted with GraphPad Prism 5. R language (*R Core Team, 2021*) was used for analysis and visualization of high throughput sequencing. Cytoscape (3.9.1) was used for visualizing the PPI network. Student's $t$-test was used to test the difference.

## RESULTS

### Characteristic features of patients

All cases were grouped by the expression level of YAP1 (low expression: $n = 626$; high expression: $n = 148$). Figures 1A–1B show the samples of two different levels of YAP1 expression in the CQMU cohort. The differences in characteristic features of the two groups were compared. Age, BMI, myometrial invasion, and PR had a $p$-value greater than 0.05 respectively, which indicated that no difference exists in these features between the two groups. However, there were significant differences in FIGO staging, cervical stromal invasion, histotypes, lymph vascular invasion, ER, Ki67, and P53 ($p$-value less than 0.05). Also, the RFS and OS were significantly different. There were 460, 58, and 108 cases in FIGO I, II, and III stages in the low expression group, respectively, while in the high expression group, 85, 16, and 47 cases were in FIGO I, II, and III stages, respectively. 96 patients had cervical stromal invasion in the low expression group, while it was 37 in the high expression group. In the low expression group, 73.6% of cases were type I, while 62.8% of cases were type I in the high expression group. In addition, 22.5% and 31.8% of cases had LVSI in low and high expression groups respectively. There were 16.1%, and 61.2% of low expressions of ER, and Ki67 respectively in the low YAP1 expression group, while the same terms were 23.6%, and 48.6% in the high YAP1 expression group. 63.6% of patients had a normal expression of P53 in the low-expression group and it was 56.8% in the high expression group. The details are shown in Table 1.

### Univariate and multivariate analyses for recurrence

We performed the univariate analysis with the common variables for recurrence in EC. The factors that have statistical significance were further analyzed by multivariate analysis. The details are shown in Table 2. These remaining factors had significant differences: myometrial invasion ($p = 0.006$), histotype ($p = 0.007$), LVSI ($p = 0.001$), ER ($p = 0.021$), Ki67 ($p = 0.017$), P53($p = 0.014$), and YAP1 ($p = 0.005$).

### High expression of YAP1 was related to poorer prognosis

We draw the Kaplan–Meier curves using the data from KM Plotter and the cohort from CQMU. The curves obtained from KM Plotter (which was shown in Figs. 1C–1D) indicated that based on mRNA expressions, 314 and 108 patients were in low and high expression groups in the KM Plotter dataset. Although there was no significant difference between the two groups in RFS and OS (RFS: log-rank $P = 0.24$; OS: log-rank $P = 0.32$), the patients in the high group were shown to have a poorer result in RFS, which was consistent with our hypothesis. Most of the recurrences of EC occurred within 3 or 5 years after surgery. It is substantial to note that patients with cancer were generally older, so there is a precipitous decline in OS after 100 months when patients may pass away due to other causes (e.g., other diseases). At the same time, there were significant differences in RFS and OS in the cohort from CQMU (which was shown in Figs. 1E–1F). The cases in the high expression group based on the results of IHC ($n = 148$) had a poorer prognosis reflected by RFS and OS (RFS: log-rank $P < 0.001$; OS: log-rank $P < 0.001$).

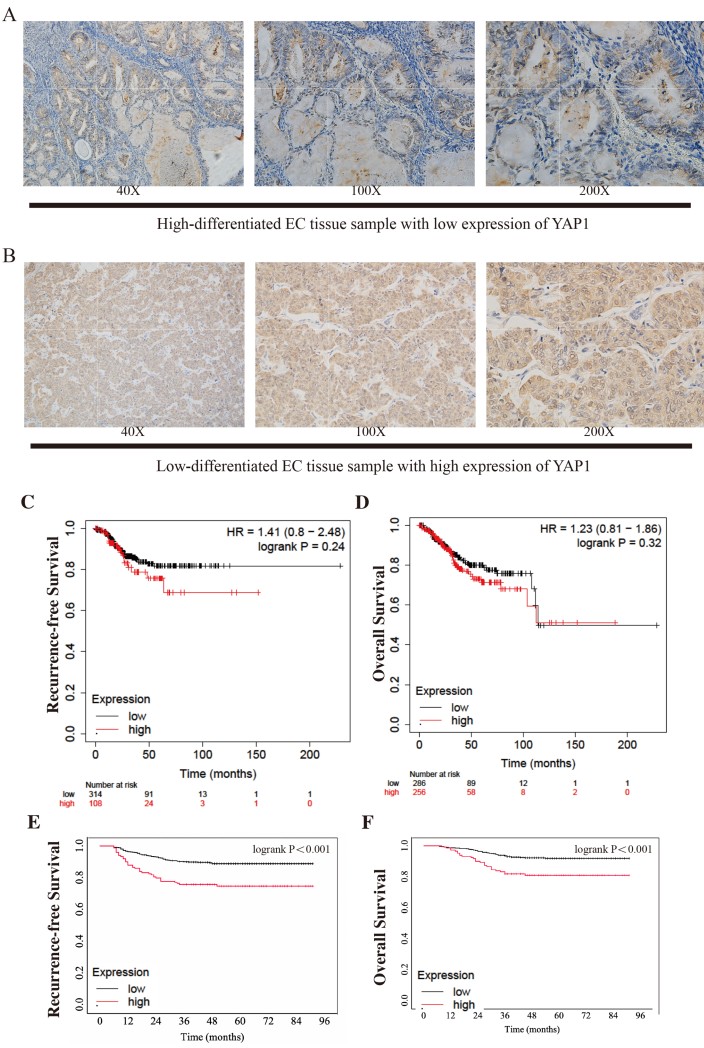

**Figure 1   Different expressions of YAP1 from CQMU cohort and Kaplan–Meier Curves from KM Plotter and CQMU cohort.** Examples of different expressions of YAP1 from CQMU cohort were displayed. (A) Case 140 (without recurrence by the end time of follow-up): High-differentiated EC with low expression of YAP1. (B) Case 81 (With recurrence by the end time of follow-up): Low-differentiated EC with high expression of YAP1. Kaplan–Meier curves show the differences of RFS and OS between high and low expression YAP1 groups. KM Plotter: low expression group, $n = 314$; high expression group, $n = 108$. CQMU cohort: low expression group, $n = 626$; high expression group, $n = 148$. Logrank was used to test the difference. (C) RFS of two groups from KM Plotter. (Logrank $P = 0.24$) (D) OS of two groups from KM Plotter. (Logrank $P = 0.32$) (E) RFS of two groups from CQMU cohort. (Logrank $P < 0.001$) (F) OS of two groups from CQMU cohort. (Logrank $P < 0.001$).

## Identification of ECSCs and detecting the expression of YAP1

As we had previously mentioned, OCT4, SOX2, NANOG, CD44, and CD133 have been widely defined as stemness-related markers (*Mirantes et al., 2013*; *Rutella et al., 2009*; *Szymonik et al., 2021*). High expression of these genes is important evidence for the identification of endometrial cancer stem cells. We cultured and isolated ECSCs from Ishikawa, HEC1-A, and RL952. We recorded the morphological changes of ECSCs by

**Table 1  Relationship between YAP1 and clinical characteristics of patients with EC.**

| Clinical characteristics | YAP1 expression (n = 774) | | P-values |
| --- | --- | --- | --- |
| | Low (n = 626) | High (n = 148) | |
| Age (years) | | | 0.315 |
| Mean ± SD | 53.70 ± 9.38 | 54.55 ± 9.99 | |
| Range | 24–81 | 34–78 | |
| BMI (kg/m$^2$) | | | 0.393 |
| Mean ± SD | 24.59 ± 3.66 | 24.88 ± 3.96 | |
| Range | 16.35–45.72 | 17.80–41.87 | |
| FIGO Staging (n, %) | | | <0.001 |
| FIGO I | 460, 73.5% | 85, 57.4% | |
| FIGO II | 58, 9.3% | 16, 10.8% | |
| FIGO III | 108, 17.3% | 47, 31.8% | |
| Cervical stromal invasion (n, %) | | | 0.005 |
| Yes | 96, 15.3% | 37, 25.0% | |
| No | 530, 84.7% | 111, 75.0% | |
| Myometrial invasion (n, %) | | | 0.213 |
| Yes | 183, 29.2% | 51, 34.5% | |
| No | 443, 70.8% | 97, 65.5% | |
| Histotypes (n, %) | | | 0.009 |
| Type I | 461, 73.6% | 93, 62.8% | |
| Type II | 165, 26.4% | 55, 37.2% | |
| LVSI (n, %) | | | 0.018 |
| Yes | 141, 22.5% | 47, 31.8% | |
| No | 485, 77.5% | 101, 68.2% | |
| ER (n, %) | | | 0.031 |
| Low | 101, 16.1% | 35, 23.6% | |
| High | 525, 83.9% | 113, 76.4% | |
| PR (n, %) | | | 0.415 |
| Low | 129, 20.6% | 35, 23.6% | |
| High | 497, 79.4% | 113, 76.4% | |
| Ki67 (n, %) | | | 0.005 |
| Low | 383, 61.2% | 72, 48.6% | |
| High | 243, 38.8% | 76, 51.4% | |
| P53 (n, %) | | | |
| Normal | 398, 63.6% | 84, 56.8% | 0.001 |
| Abnormal | 228, 36.4% | 64, 43.2% | |
| Recurrence (n, %) | | | <0.001 |
| Yes | 68, 10.9% | 37, 25% | |
| No | 558, 89.1% | 111, 75% | |
| Recurrence-free survival (months) | | | 0.359 |
| Median | 49 | 49 | |
| Range | 6–91 | 6–91 | |

**Table 1** (*continued*)

| Clinical characteristics | YAP1 expression ($n = 774$) | | P-values |
|---|---|---|---|
| | Low ($n = 626$) | High ($n = 148$) | |
| Death ($n$, %) | | | <0.001 |
| Yes | 48, 7.7% | 28, 18.9% | |
| No | 578, 92.3% | 120, 81.1% | |
| Overall survival (months) | | | 0.653 |
| Median | 51 | 51.5 | |
| Range | 7–91 | 8–91 | |

**Notes.**
YAP1, Yes-associated protein 1; FIGO, Federation International of Gynecology and Obstetrics; BMI, body mass index; LVSI, lymph vascular invasion; SD, standard deviation; ER, estrogen receptor; PR, progesterone receptor.

**Table 2  Univariate and multivariate analyses for recurrence.**

| Clinicopathological | Univariate analysis | | | Multivariate analysis | | |
|---|---|---|---|---|---|---|
| | HR | 95%CI | p Value | HR | 95%CI | p Value |
| Age (ref. <60) | 2.441 | 1.663–3.584 | <0.001 | 1.453 | 0.974–2.166 | 0.067 |
| FIGO | | | <0.001 | | | 0.093 |
| FIGO II (ref. FIGO I) | 2.613 | 1.356–5.035 | 0.004 | 1.400 | 0.620–3.161 | 0.418 |
| FIGO III (ref. FIGO I) | 7.296 | 4.793–11.107 | <0.001 | 1.804 | 1.054–3.086 | 0.031 |
| Cervical stromal invasion (ref. No) | 2.973 | 1.991–4.438 | <0.001 | 1.215 | 0.727–2.031 | 0.458 |
| Myometrial invasion (ref. No) | 3.792 | 2.569–5.597 | <0.001 | 1.824 | 1.191–2.794 | 0.006 |
| Histotypes (ref. Type I) | 5.866 | 3.908–8.805 | <0.001 | 1.959 | 1.200–3.198 | 0.007 |
| LVSI (ref. No) | 4.650 | 3.161–6.841 | <0.001 | 2.070 | 1.336–3.207 | 0.001 |
| ER (ref. High) | 6.139 | 4.184–9.008 | <0.001 | 2.023 | 1.110–3.686 | 0.021 |
| PR (ref. High) | 4.618 | 3.148–6.774 | <0.001 | 1.045 | 0.575–1.897 | 0.886 |
| Ki67 (ref. Low) | 2.966 | 1.982–4.438 | <0.001 | 1.676 | 1.098–2.566 | 0.017 |
| P53 (ref. Normal) | 1.903 | 1.298–2.791 | 0.001 | 1.643 | 1.108–2.437 | 0.014 |
| YAP1 (ref. Low) | 2.516 | 1.686–3.755 | <0.001 | 1.803 | 1.190–2.731 | 0.005 |

**Notes.**
YAP1, Yes-associated protein 1; FIGO, Federation International of Gynecology and Obstetrics; BMI, body mass index; LVSI, lymph vascular invasion; ER, estrogen receptor; PR, progesterone receptor; CI, confidence interval; HR, hazard ratio.

using microscopy on the third, fifth, and seventh days, which are shown in Fig. 2A. It could be seen that the cells gradually formed spheres starting from the third day. Among them, Ishikawa and HEC1-A show the highest efficiency in spheric formation and proliferation, which is related to the most stem cell-like state. Therefore, we used these two cell lines for the subsequent experiments. To further validate the ECSCs that we isolated, bulky ECSCs spheroids were collected on the eighth day. We examined the molecules associated with ECSCs as previously mentioned. As hypothesized, qRT-PCR suggested that mRNA levels of SOX2, OCT4, CD44, CD133, and NANOG, the stemness-related molecules as defined, were significantly increased in ECSCs$^{Ishikawa}$ and ECSCs$^{HEC1-A}$, compared with its original cell lines. The results are shown in Figs. 2B–2C. Subsequently, we further confirmed the elevated expression of these molecules at the protein level by western blot assay, which are shown in Figs. 2D–2E. These results

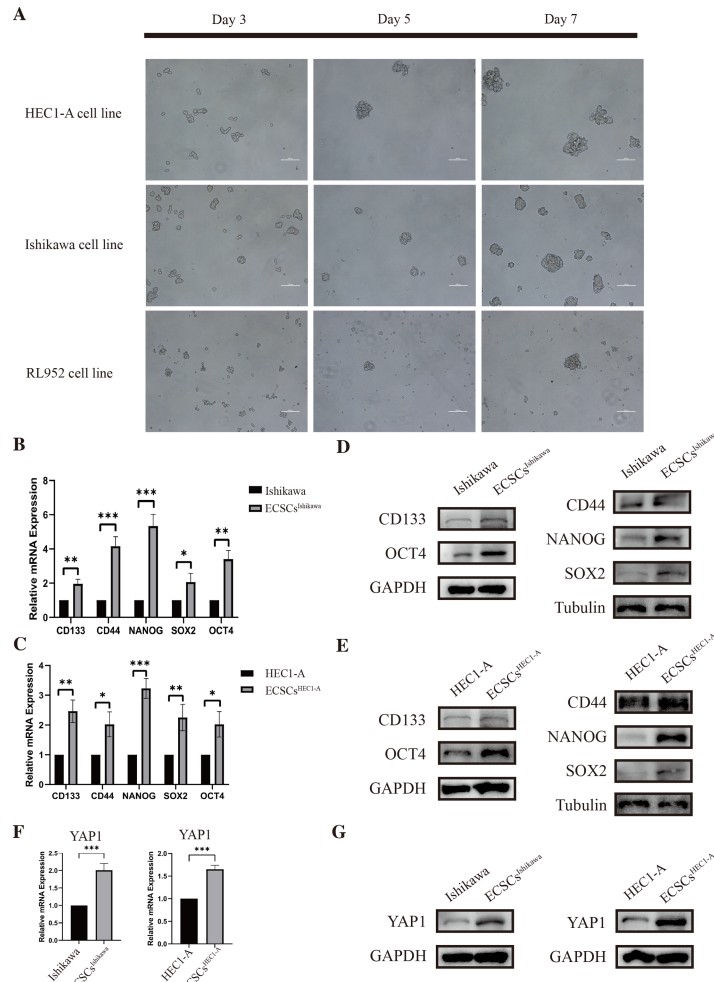

**Figure 2** **Identification of ECSCs and detecting expression of YAP1.** HEC1-A, Ishikawa, and RL952 were used for culture of ECSCs. The most stem-like state cell lines were HEC1-A and Ishikawa. These two cell lines were used for the subsequent experiments. Student's $t$-test was used to test the differences between two groups. (A) The spheric formation images of HEC1-A, Ishikawa, and RL952 on the 3rd, 5th and 7th day were displayed (100X). (B–C) Analysis of the relative mRNA expressions of SOX2, OCT4, NANOG, CD44, and CD133 in different ECSCs, compared with its original cell lines. (qRT-PCR were tested for three times. Mean ± SD. $^*P < 0.05$, $^{**}P < 0.01$, $^{***} P < 0.001$.) (D–E) Protein expressions of SOX2, OCT4, NANOG, CD44, and CD133 by western blot in different ECSCs and controls. (F) Relative mRNA expression of YAP1 detected by qRT-PCR. (qRT-PCR were tested for three times. Mean ± SD. $^{***}$ $P < 0.001$) (G) Protein expression of YAP1 in different ECSCs, compared with its original cell lines.

confirmed that the stem-like cells we cultured and isolated on the eighth day were ECSCs, which had the characteristics of ECSCs.

Also, the examination of the expression of YAP1 in EC cell lines and ECSCs was performed at the same time. The expression of YAP1 in ECSCs$^{\text{Ishikawa}}$ and ECSCs$^{\text{HEC1−A}}$ showed a significant increase compared with Ishikawa and HEC1-A respectively, which is confirmed by qRT-PCR and western blot assay (the results are shown in Figs. 2F–2G). The result was consistent with our previous speculation.

## Down-regulation of YAP1 suppressed the stemness in Ishikawa and HEC1-A

The changes in stemness of Ishikawa and HEC1-A were evaluated after YAP1 knockdown by siRNA. The knockdown of YAP1 expression was confirmed by qRT-PCR, which was shown in Figs. 3A–3B. Then, we further cultured the cells after YAP1 expression knockdown in the stem cell medium as described previously. After YAP1 knockdown, the cells still had the ability of spheric formation (same shape as ECSCs). However, it showed a reduced trend of sphere-forming in number and size. Figures 3C–3F show the changes in the sphere-forming ability of stem cells in different groups on the seventh day. The number of spheres in the siRNA#1 and siRNA#2 groups was less compared with the control group. The counts of spheres were recorded in File S4.

Also, we tested the expression of stemness molecules 48 h after YAP1 knockdown. Significant decreases in SOX2, OCT4, CD44, CD133, and NANOG were observed in qRT-PCR. Western blot also demonstrated the same trend. The graphs were displayed in Figs. 3G–3J. This result confirmed that down-regulation of YAP1 would suppress the stemness in Ishikawa and HEC1-A.

In addition, we randomly selected 20 patients for immunohistochemical staining of YAP1 and its target genes (CTGF, CYR61, ANKRD1, AMOTL2), and TAZ. The aim was to explore the association between these genes. The results were recorded in File S5. We also detected changes in the target genes (CTGF, CYR61, ANKRD1, AMOTL2) of YAP1 by qRT-PCR in YAP1 knockdown Ishikawa and HEC1-A cells, and we obtained similar results to other studies. Similar results were obtained by knockdown of TEAD-1 and the use of YAP-TEAD1 inhibitor (Peptide17). The results are shown in File S6.

## High-throughput sequencing of transcriptome revealed the potential pathways in ECSCs

The column graph and volcanic graph were displayed in Figs. 4A–4B, which indicated that 2,675 genes were significantly increased in ECSCs$^{Ishikawa}$ (cultured in DMEM/F12 with 10 ng/ml hFGF, 20 ng/ml hEGF, and B27 supplement) compared with Ishikawa (cultured in DMEM with 10% FBS), while 253 genes were decreased. The heatmap in Fig. 4C shows the top 100 genes with the most significant changes, which included YAP1 (the details are shown in File S7). Gene ontology (GO) enrichment analysis indicated that the most related pathways were plasma membrane, membrane, integral component of the plasma membrane, integral component of membrane, extracellular space, extracellular region, extracellular matrix organization, extracellular matrix, and cell surface. Kyoto Encyclopedia of Genes and Genomes (KEGG) pathway enrichment indicated that typical pathways in cancer were involved, including the Wnt signaling pathway, and the p53 signaling pathway. Also, the HIPPO signaling pathway had a significant difference between ECSCs$^{Ishikawa}$ and Ishikawa ($p$-value $= 0.000227292$). The details are shown in Figs. 4D–4E.

## Protein–protein interactions network was constructed

Based on the previous result of high throughput sequencing, the protein-protein interactions network was constructed by STRING. The hub genes (top 100) were displayed in Fig. 5A (the details are shown in File S8). We further analyzed the core molecules related

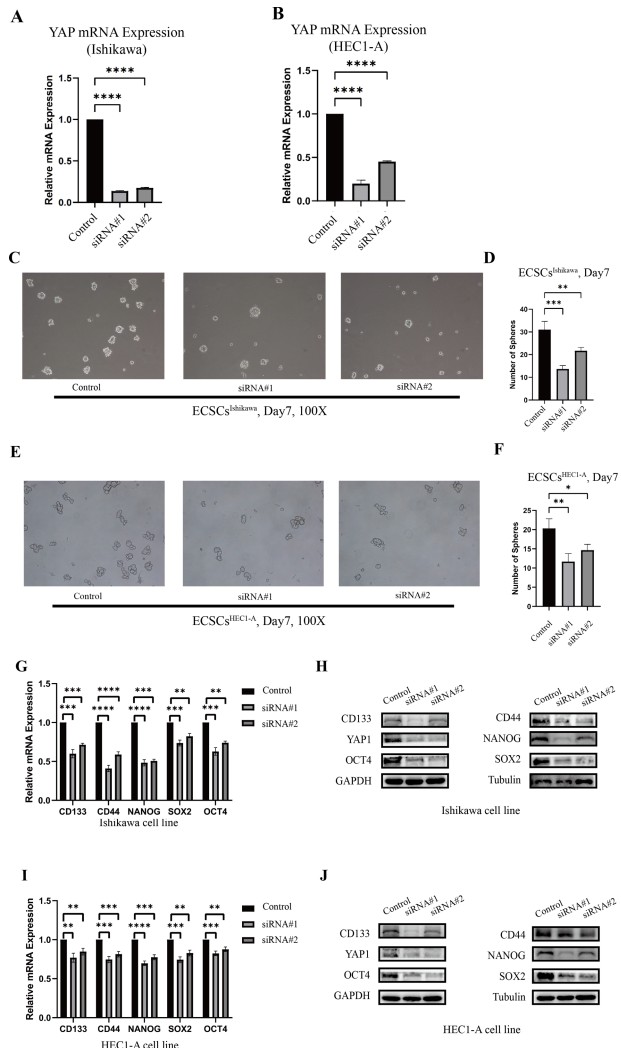

**Figure 3** **The different abilities of sphere formation after YAP1 knockdown and expressions of SOX2, OCT4, NANOG, CD44, and CD133 after YAP1 knockdown.** The efficiency of spheres formation was compared after YAP1 knockdown. (A–B) Down-regulation of YAP1 by siRNA was confirmed by qRT-PCR. (qRT-PCR were tested for three times) (C) Images of spheres formation of ECSCs[Ishikawa] after YAP1 knockdown on the 7th day (100X). (D) Counting of ECSCs[Ishikawa] spheres. Mean number of five random scopes stood for the ultimate result. (E) Images of spheres formation of ECSCs[HEC1-A] after YAP1 knockdown on the 7th day (100X). (F) Counting of ECSCs[HEC1-A] spheres. Mean number of five random scopes stood for the ultimate result. After YAP1 knockdown in Ishikawa and HEC1-A, the expressions of SOX2, OCT4, NANOG, CD44, and CD133 were detected by qRT-PCR and western blot. (G) mRNA expressions of SOX2, OCT4, NANOG, CD44, and CD133 in Ishikawa were detected by qRT-PCR. (qRT-PCR were tested for three times. Student's $t$-test was used to test the differences between two groups. Mean $\pm$ SD. *$P < 0.05$, **$P < 0.01$, *** $P < 0.001$, ****$P < 0.0001$.) (H) Protein expressions of YAP1, SOX2, OCT4, NANOG, CD44, and CD133 in Ishikawa were detected. (I) mRNA expressions of SOX2, OCT4, NANOG, CD44, and CD133 in HEC1-A were detected by qRT-PCR. (qRT-PCR were tested for three times. Student's $t$-test was used to test the differences between two groups. Mean $\pm$ SD. *$P < 0.05$, **$P < 0.01$, *** $P < 0.001$, ****$P < 0.0001$.) (J) Protein expressions of YAP1, SOX2, OCT4, NANOG, CD44, and CD133 in HEC1-A were detected.

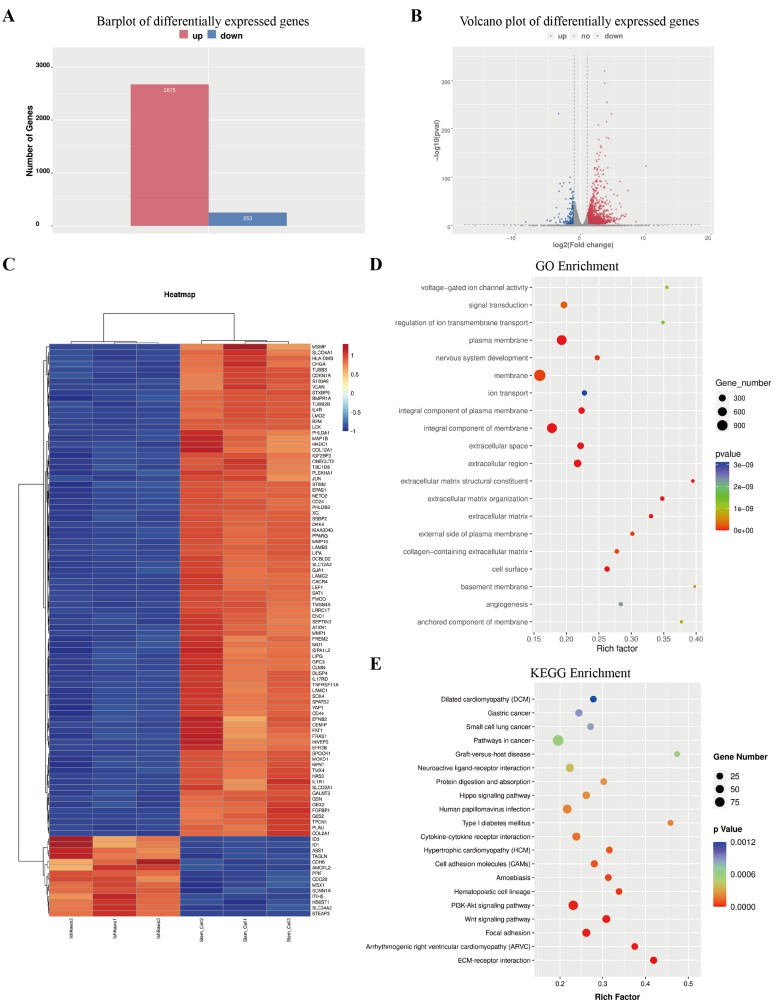

**Figure 4** **Outcome of high through-put sequencing.** High through-put sequencing was performed by using total RNA of ECSCs$^{Ishikawa}$, compared with Ishikawa cell line. (A–B) Plots indicated that 2,675 genes were up-regulated in ECSCs$^{Ishikawa}$ while 253 genes were down-regulated. log2 (fold change) $\geq 1$ or log2 (fold change) $\leq -1$ and $p$ value $< 0.05$. (C) Heatmap of the top 100 significant genes. (D) GO enrichment analysis. (E) KEGG enrichment analysis.

to YAP1, which are shown in Fig. 5B. These genes had close correlation with YAP1 based on our result: LEF1, FAT1, JUN, AMOTL2, DKK4, GJA1.

## DISCUSSION

Many studies started to focus on the role that cancer stem-like cells played in recurrence, drug resistance, and metastasis (*Huang et al., 2020*; *Najafi, Mortezaee & Majidpoor, 2019*). In our study, we focused on the area of ECSCs which remained unclear. After grouping 774 patients by their expression levels of YAP1, we found a significant difference in prognosis between high and low expression groups. The data from KM Plotter also confirmed the same trend. Subsequently, we explored the underlying mechanism behind it and found that YAP1, a core molecule of the HIPPO pathway, played an important role in the
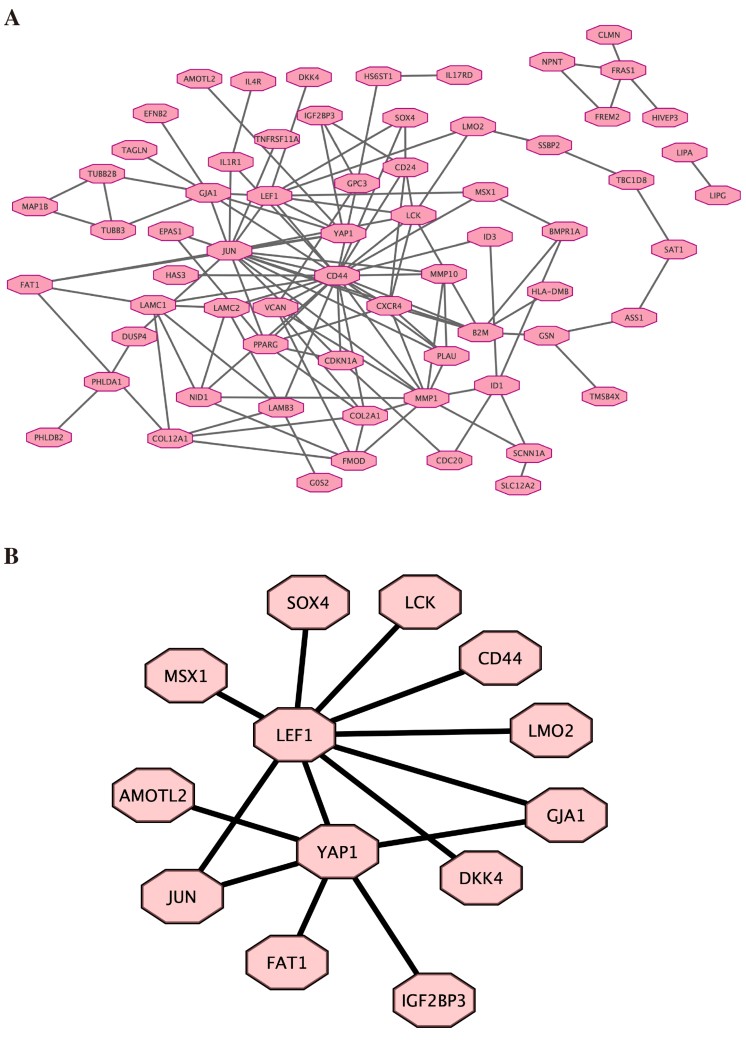

A

B

**Figure 5** **Protein–protein interactions network.** Protein-protein interactions (PPI) network was constructed by STRING. (A) PPI network. (B) PPI network related to YAP1.

development and maintenance of CSCs' stemness (*Ma et al., 2019*). Many studies have also confirmed this view. After the down-regulation of YAP1 *in vitro*, the stemness of Ishikawa and HEC1-A was significantly inhibited. This confirmed our view that YAP1 may influence the prognosis of patients with EC by regulating stemness.

First, YAP1 had now been shown to play a key role in many types of tumors. Researchers had illustrated the function of YAP1 as a prognosis marker in advanced gallbladder cancer (*García et al., 2020*). Overexpression of YAP1 was also proved to be associated with poorer prognosis in breast cancer (*Guo et al., 2019*). Our study also drew a similar result: patients with EC who had a high expression level of YAP1 tended to have a poorer prognosis. In parallel, we found that immunohistochemical staining suggests that patients with different expressions of YAP1 have different prognoses. This suggests the potential of YAP1 as an immunohistochemical index in the prediction of recurrence. Especially in developing

countries, immunohistochemistry can be applied to more patients conveniently and economically. Our results suggested that YAP1 was a promising molecule for predicting the prognosis of patients with EC and was also a potential target for relapse prevention therapy, which was certainly meaningful and interesting.

Secondly, studies have suggested YAP1 has a strong impact on the development and maintaining stemness in many cancers. YAP1 overexpression contributes to the induction of cancer stemness in prostate cancer (*Lee et al., 2021*). Therefore, in this study, we also focused on whether YAP1 has an influence on the stemness of EC cell lines. As it's expected, the knockdown of YAP1 *in vitro* significantly suppressed the stemness of Ishikawa and HEC1-A, which revealed a potential mechanism for YAP1 to affect the prognosis of EC patients, by regulating the stemness of EC. Further study of the role that YAP1 acts in stemness will provide the basis for subsequent drug development, *etc.*

Third, the existing studies were not well understood for ECSCs, although the important role of several pathways (such as Wnt pathway, NOTCH pathway (*Götte et al., 2011*; *Wang et al., 2017*) in ECSCs had been proposed. The pathways behind ECSCs have not been systematically described. Therefore, we performed high-throughput transcriptome sequencing of ECSCs. YAP1, as well as the HIPPO pathway, were proven to be crucial in regulating stemness. Our results might provide new insight for others to further investigate ECSCs. More experiments should focus on the role of the HIPPO pathway played in ECSCs. Notably, we found that there are still many genes in stem cells that differ from normal cell lines, and these need to be further investigated by scholars.

Finally, by high-throughput sequencing, we found that these genes (LEF1, FAT1, JUN, AMOTL2, DKK4, and GJA1) were among the top 100 genes and were associated with YAP1. Decreasing FAT1 expression significantly activated the YAP1 signaling pathway and promoted collective invasion. A study also revealed that YAP1/ JUN/stemness axis may play an important role in iron metabolism (*Zhou et al., 2023*). AMOTL2 can interact with both YAP1 and LEF1 to control tissue size in zebrafish (*Agarwala et al., 2015*). YAP1 and GJA1 act as related hub genes in Alzheimer's disease (*Wang & Wang, 2020*). These studies support the reliability of the results we obtained in high-throughput sequencing. However, the specific molecular mechanisms behind it still need more research.

There were a few limitations in our research: (1) This study was a single-center study, and the sample size was limited by the follow-up time. More cases were needed to get more accurate results. (2) Although our results were sufficient to confirm the role YAP1 played in ECSCs, more experiments *in vivo* would furtherly confirm our findings. (3) The potential pathways and molecules in ECSCs were revealed by high through-put sequencing, but they should be verified in the following studies.

## CONCLUSIONS

Our study indicated that YAP1 affected the prognosis of patients with EC by regulating stemness. High expression of YAP1 resulted in poorer outcomes. Knockdown of YAP1 would significantly inhibit the stemness of Ishikawa and HEC1-A. HIPPO pathway had a strong impact on ECSCs. It can be concluded that YAP1 is a promising biomarker for

predicting the prognosis of EC and is a potential molecule for targeting the recurrence and metastasis of EC.

### Abbreviations

| | |
|---|---|
| **EC** | endometrial cancer |
| **CQMU** | Chongqing Medical University |
| **CSCs** | cancer stem-like cells |
| **ECSCs** | endometrial cancer stem-like cells |
| **YAP1** | Yes-associated protein 1 |
| **FIGO** | Federation International of Gynecology and Obstetrics |
| **BMI** | body mass index |
| **LVSI** | lymph vascular invasion |
| **RFS** | recurrence-free survival |
| **OS** | overall survival |
| **qRT-PCR** | quantitative reverse transcription PCR |
| **WB** | western blot |
| **hEGF** | human epidermal growth factor |
| **hFGF** | human fibroblast growth factor |
| **SOX2** | SRY-box containing gene 2 |
| **OCT4** | octamer-binding transcription factor 4 |
| **CD44** | cluster of differentiation-44 |
| **CD133** | cluster of differentiation-133 |

### Funding
The authors received no funding for this work.

### Competing Interests
The authors declare there are no competing interests.

### Author Contributions

- Wei Kong conceived and designed the experiments, performed the experiments, prepared figures and/or tables, authored or reviewed drafts of the article, and approved the final draft.
- Yuzhen Huang conceived and designed the experiments, performed the experiments, analyzed the data, prepared figures and/or tables, authored or reviewed drafts of the article, and approved the final draft.
- Peng Jiang conceived and designed the experiments, performed the experiments, analyzed the data, prepared figures and/or tables, authored or reviewed drafts of the article, and approved the final draft.
- Yuan Tu conceived and designed the experiments, performed the experiments, prepared figures and/or tables, and approved the final draft.
- Ning Li performed the experiments, authored or reviewed drafts of the article, and approved the final draft.
- Jinyu Wang performed the experiments, prepared figures and/or tables, authored or reviewed drafts of the article, and approved the final draft.
- Qian Zhou performed the experiments, authored or reviewed drafts of the article, and approved the final draft.
- Yunfeng Zheng performed the experiments, analyzed the data, prepared figures and/or tables, authored or reviewed drafts of the article, and approved the final draft.
- Shikai Gou performed the experiments, authored or reviewed drafts of the article, and approved the final draft.
- Chenfan Tian performed the experiments, analyzed the data, prepared figures and/or tables, authored or reviewed drafts of the article, and approved the final draft.
- Rui Yuan conceived and designed the experiments, analyzed the data, prepared figures and/or tables, authored or reviewed drafts of the article, and approved the final draft.

## Human Ethics

The following information was supplied relating to ethical approvals (i.e., approving body and any reference numbers):

All procedures were performed in accordance with relevant guidelines and were in line with World Medical Association Declaration of Helsinki. This study had been approved by the Ethics Committee of the First Affiliated Hospital of Chongqing Medical University (2021-676).

## Data Availability

Raw data are available in the Supplemental Files.

## Supplemental Information

Supplemental information for this article can be found online at http://dx.doi.org/10.7717/peerj.15891#supplemental-information.

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
