# Peer review of "YAP1 affects the prognosis through the regulation of stemness in endometrial cancer"

_PeerJ, doi:10.7717/peerj.15891_

## Round 0.1 · original submission · Major Revisions

Please, address the reviewers' comments. There are consensual doubts and issues with methodology and results (and their interpretation) that should be addressed. Many thanks for your submission.

Reviewer 1 ·

Basic reporting

In this article, Kong and colleagues investigate the influence of high YAP staining/expression in Endometrial cancer. They show that tumors with high YAP staining tend to be more aggressive and have a shorter RFS and OS (at least in their cohort). In addition, they show that cultured tumor (stem) cells have higher YAP expression and knockdown of YAP decreases the expression of several stem cell markers. All in all, these results are interesting, but several points need to be addressed.

- This article would greatly benefit from additional editing of a fluent English speaker. It is very hard to follow the authors’ narrative
- Is the YAP staining nuclear? Can the authors quantify this?
- Expression analysis of YAP target genes (such as CTGF, CYR61, ANKRD1, AMOTL2)
- Was the survival analysis (OS/RFS) (KM Plotter and CQMU) based on RNA expression or IHC staining of YAP? Is this the same between both platforms?
- Is there a correlation between the expression of YAP (based on RNA) and the intensity of YAP histochemistry staining (protein)?
- Is there a correlation between YAP expression (or IHC staining) and the expression of YAP target genes (RNA)?
- Is the staining for TAZ similarly elevated in the YAP1-high group?
- Do the authors know anything about inactivating mutations in Hippo pathway genes (regulating YAP activity) in the YAP high tumors?
- Section 2.5: was the knockdown of YAP accompanied with a reduction in the expression of YAP target genes (such as CTGF, CYR61, ANKRD1, AMOTL2)? The authors should add additional experiments: what happens when TEAD transcription factors (TEAD1-4) are knocked down? Does this lead to a similar effect? Does treatment with a YAP1-TEAD inhibitor also lead to similar effects?
- Section 2.6: The description of the performed experiments is very rudimentary, and it would greatly benefit the reader if this could be written and explained clearer. Eg it was unclear to me (until I check it in the methods section) that the paternal Ishikawa cell line was cultured in serum-containing medium. Also, it should be clearly explained/labeled that the 2675 up-regulated genes are up-regulated in the serum-free stem cell condition.. etc.

Experimental design

See above

Validity of the findings

see above

Additional comments

see above

Reviewer 2 ·

Basic reporting

The authors have few English grammar errors but may need to improve their expressions for better understanding. The author followed a good article structure with a detailed explanation of data collection and methodology. They attached their raw data to the supplementary files. Figures need to have a higher resolution and a larger font on their axis and label. I suggest that they can use vector graphics files like SVG rather than raster graphics like PNG/BMP. I have some suggestions about the motivation of the research in the general comments.

Experimental design

This research falls within the scope of PeerJ. They have a detailed description of the methodology. They may need to narrow down the research problem and further explain how this research will fill the knowledge gap.

The authors provided the ethical approval statement in Chinese. They should attach a translated version to the paper in the revision for the general international audience. No personal identifying information was provided in the data. They confirmed that written consent had been obtained from patients. Overall, their work meets the ethical requirements of PeerJ.

Validity of the findings

The statistical significance of the data was assessed in the article. Conclusions were linked with their experimental results. This paper reveals novel connections between YAP expression and ECSC stemness. I have several suggestions about the research impact in the general comments.

Additional comments

Overall, I suggest accepting this paper with a revision. In general, they need to provide more description, explanation, and discussion in the result section.
Several aspects of this work could be improved:
1. The introduction needs more evidence to support the motivation for linking the HIPPO pathway to endometrial cancer and cell stemness. Between line 79 to line 82, they also need to justify the purpose of focusing on ECSC stemness rather than an overview of EVSV biomarkers. It is well-known that YAP1 and the HIPPO pathway control cell apoptosis and stem cell renewal. Between line 83 to line 97, the authors mentioned multiple citations about general cancer-related signaling pathways. I think they should focus on why HIPPO pathways will provide valuable biomarkers targeting endometrial cancer rather than other pathways.
2. In section 2.1, I do see a significant difference in the survival rate of the patients. I suggest they further explain why the overall survival rate shows no difference after 100 months, but the recurrence-free survival rate shows a significant difference.
3. In sections 1.4 and 2.4, they describe the sphericity of the cell line. I suggest that they further describe the software used to measure the sphericity of the cell line.
4. In section 2.4, I suggest that they explain why SOX2 and other biomarkers are related to stemness with reference before talking about the results. It is important to provide background information for the reader to understand the experiment and justify the motivation.
5. In sections 2.6 and 2.7, I suggest that they further discuss the potential impact of those up/downregulated genes on the ECSC cell.
6. In the discussion section, I suggest that they emphasize the novelty and impact of this paper. Rather than comparing it with other research, they should stress how their data will narrow down the knowledge gap and provide new insights for readers.
7. I suggest they save the data file to CSV files rather than SAV files. The SAV file can only be opened with SPSS software which provides potential inconvenience for readers.

Annotated reviews are not available for download in order to protect the identity of reviewers who chose to remain anonymous.

Reviewer 3 ·

Basic reporting

Peer J #81171

YAP1 affects the prognosis through the regulation of stemness in endometrial cancer by Wei Kong et al.

In the present study, the author explored the role of YAP1, a key component of the HIPPO pathway, on the stemness of endometrial cancer and revealed its influence on prognosis in clinical samples as well as in cell lines. The authors investigated the expression of the stem cell markers (SOX2, OCT4, CD44, CD133 and NANOG) at mRNA and protein levels in endometrial cancer cells and their spheroids. They claimed that YAPl knockdown reduced the sphere number and size as well as the expression of stem cell markers at the mRNA and protein levels.
It is clear from this and previous work that YAP1 plays an important role in the malignancy of endometrial cancer. Thus, it remains important to decipher the molecular mechanisms behind the stemness of endometrial malignancy as they do in this manuscript.
The authors have produced good-quality data, but English language needs extensive revision. The authors did not check spelling carefully during manuscript writing. I have made specific comments,

Introduction

Lane 65: …“Increase of aged population” …….should be… “An increase in the aged population”…..

Lane 74: Correct the sentence.
“Also, reported by another research, CD44 were related to EC stemness.”

Lane 78: ….“deregulation”…..should be…. “the deregulation”…

Lane 79: Correct the sentence.
“These studies had provided essential supports for the research of ECSCs.”

Lane 92: …. “Similar”….should be… “A similar” …..

Lane 98: Correct the sentence.

“Therefore, the aims of our study were to investigate the effect of YAPl in the prognosis of patients with EC, to verify the function of YAPl on stemness of EC cell lines in vitro, and to explore the potential intermolecular interactions of YAPl by high-throughput sequencing.”

Material and Methods

Lane 117: ….“follow”….should be ….“ follows”…

Lane 118: ….“follow-up”…should be …“the follow-up”…

Lane 128:…..“of Chongqing Medical University”…..should be “at Chongqing Medical University”….

Results

Lane 229: ….“stage in low expression group”….should be …“stages in the low expression group”…

Lane 240: ….“univariate”…..should be ….“the univariate”…..

Lane 242: “Difference”…..should be.. “differences”…

Lane 270: …“were”…should be …“was”…

Experimental design

The authors experimental design is good and produced good-quality data. Author should provide detailed for counting the number of spheres and comparision between YAP1 WT and knockdown cells.

The author must provide a detailed statistical analysis for each figure.

Validity of the findings

The authors produced good-quality data but overinterpreted the results for sphere-forming ability in YAP1 WT and knockdown cells (Figure 3C-E). The main problem is that they did not explain how they count the number of spheres. Because as with time spheres attach to each other and grow in size/no. of cells per sphere. Therefore, it will be difficult to count the number of spheres or no. of cells in one sphere.
Authors must be explain in detailed for it.

---

## Round 0.2 · Major Revisions

Dear authors,

Thank you for submitting your revised manuscript. However, I regret to inform you that I still find it necessary to request major changes. While I appreciate your efforts to improve the presentation and description of the data, I have observed that the significant concerns raised by the reviewers have not been adequately addressed. I strongly advise you to carefully review the reviewers' comments once again and address them thoroughly, particularly in the rebuttal letter.

It is crucial to remember that certain aspects discussed in the rebuttal letter should also be incorporated into the manuscript itself, allowing the readers to access this information. Many of the points raised by the reviewers are valid and are likely to be questions that researchers in the field will have. Moreover, providing sufficient details regarding data analysis, cell sphericity measurements, and highlighting the importance of your work are essential for ensuring the validity of your research. These aspects should not be overlooked, downplayed, or dismissed.

Lastly, it is important to note that adding references without providing background information or explanations may be deemed inappropriate in certain situations and could be considered malpractice. Therefore, I urge you to thoroughly review both recent and past reviewer comments and address them diligently.

Reviewer 1 ·

Basic reporting

The authors have replied to most of my concerns, however, a few minor issues remain. Please see below.

Original comment: [2]- Is the YAP staining nuclear? Can the authors quantify this?
New comment: I am sorry if I did not articulate my original comment clear enough – but I was asking the authors to quantify their IHC stainings to assess if YAP is nuclear in these tumors. Please add these quantifications.

Original comment: [3]- Expression analysis of YAP target genes (such as CTGF, CYR61, ANKRD1, AMOTL2)
New comment: It is great that the authors added these correlations, however, the .svg file that was supplied as Suppl. Figure 4 is missing some of the embedded .png files and the figure panels are not displayed/are missing. Could the authors please correct this?

Original comment: [9]- Section 2.5: was the knockdown of YAP accompanied with a reduction in the expression of YAP target genes (such as CTGF, CYR61, ANKRD1, AMOTL2)? The authors should add additional experiments: what happens when TEAD transcription factors (TEAD1-4) are knocked down? Does this lead to a similar effect? Does treatment with a YAP1-TEAD inhibitor also lead to similar effects?
New comment: It is great that the authors did these new analysis and the described results are promising. However, I can’t find the mentioned figures/graphs in Suppl Fig 6. This file only contains the protein-protein interaction network. Could the authors please supply the correct figures?

Experimental design

see above

Validity of the findings

see above

Additional comments

see above

Reviewer 2 ·

Basic reporting

The authors improved some expression in the paper. The author followed a good article structure with a detailed explanation of data collection and methodology. They attached their raw data to the supplementary files. After the revision, they simply added many references without talking about them in detail. So as a reader, we do not know how the field is progressed and why their discovery is important. The quality of the manuscript barely improved after the revision.

Experimental design

They added several tables to the rebuttal letter but failed to add these results to the main text. They failed to describe their results in detail before the revision and things are the same after the revision in many places. For example, several reviewers asked about the sphere formation of the cell line. Even though they added a reference to that part, they still failed to provide their raw data or how they did the statistical analysis. In the reference they cited, there is more detail about the whole experiment across multiple passages.

Validity of the findings

They did not provide enough description of their findings. For example in 2.7.

2.7 Protein-protein interactions network was constructed
Based on the previous result of high throughput sequencing, the protein-protein interactions network was constructed by STRING. The hub genes were displayed in Figure 5A (The details were shown in supplementary file 5).Supplementary File 6). We further analyzed the core molecules related to YAP1, which were shown in Figure 5B. These genes had close correlation with YAP1 based on our result: LEF1, FAT1, JUN, AMOTL2, DKK4, GJA1.

There are only 3 sentences describing important NGS data. Without talking about why those genes may be related to YAP1 expression and what this result indicates or any further application.

Reviewer 3 ·

Basic reporting

Peer J #81171-v1

This manuscript is a re-submission of a manuscript I reviewed 3 months ago. At that time, I was supportive and interested in the author's claims (that YAP1 affects the prognosis through the regulation of stemness in endometrial cancer), but simultaneously was very underwhelmed by the actual experimental data provided to support that claim. I was fully expecting a vastly improved manuscript after revision. In the author's defense, all the experimental data has been improved by providing additional data, rearranging, and explanation. Now provided data are more convincing to support that claim. The authors addressed all the criticisms. Therefore, I would like to suggest this manuscript to accept for publication after the editor's decision.

Experimental design

no comment.

Validity of the findings

no comment.

Additional comments

no comment.

---

## Round 0.3 · accepted · Accept

Dear authors, thank you for your work and efforts. I believe your work is now suitable for publication. Many congratulations! Please, refer to the reviewers notes for further comments.

Reviewer 1 ·

Basic reporting

The authors have addressed my previous comments and concerns.

Experimental design

see above

Validity of the findings

see above

Additional comments

see above

Reviewer 2 ·

Basic reporting

The revision met the publication requirement of PeerJ with enough references, plenty of professional figures, and good English expression.

Experimental design

The experiment design is reasonable and the authors provide enough data to show the experimental result.

Validity of the findings

They discussed the result of the experiments separately. However, these descriptions are usually explicit descriptions of the data rather than forming a conclusion or a question for the next section/experiment to discuss. It is good that they added more content in the discussion to reveal the importance of the paper in the discussion section.

Additional comments

I suggest this paper be published as it is or with minor revisions after the editor's decision.